# *Clostridioides difficile* Toxins: Host Cell Interactions and Their Role in Disease Pathogenesis

**DOI:** 10.3390/toxins16060241

**Published:** 2024-05-24

**Authors:** Md Zahidul Alam, Rajat Madan

**Affiliations:** 1Department of Pathology and Laboratory Medicine, Brody School of Medicine, East Carolina University, 600 Moye Boulevard, Greenville, NC 27858, USA; 2Division of Infectious Diseases, Department of Internal Medicine, University of Cincinnati College of Medicine, Cincinnati, OH 45267, USA; madanrt@ucmail.uc.edu; 3Veterans Affairs Medical Center, Cincinnati, OH 45220, USA

**Keywords:** *Clostridioides difficile* infection, *C. difficile* transferase (CDT), toxin A (TcdA) and toxin B (TcdB), GTPase proteins, pathogenicity locus (PaLoc)

## Abstract

*Clostridioides difficile*, a Gram-positive anaerobic bacterium, is the leading cause of hospital-acquired antibiotic-associated diarrhea worldwide. The severity of *C. difficile* infection (CDI) varies, ranging from mild diarrhea to life-threatening conditions such as pseudomembranous colitis and toxic megacolon. Central to the pathogenesis of the infection are toxins produced by *C. difficile*, with toxin A (TcdA) and toxin B (TcdB) as the main virulence factors. Additionally, some strains produce a third toxin known as *C. difficile* transferase (CDT). Toxins damage the colonic epithelium, initiating a cascade of cellular events that lead to inflammation, fluid secretion, and further tissue damage within the colon. Mechanistically, the toxins bind to cell surface receptors, internalize, and then inactivate GTPase proteins, disrupting the organization of the cytoskeleton and affecting various Rho-dependent cellular processes. This results in a loss of epithelial barrier functions and the induction of cell death. The third toxin, CDT, however, functions as a binary actin-ADP-ribosylating toxin, causing actin depolymerization and inducing the formation of microtubule-based protrusions. In this review, we summarize our current understanding of the interaction between *C. difficile* toxins and host cells, elucidating the functional consequences of their actions. Furthermore, we will outline how this knowledge forms the basis for developing innovative, toxin-based strategies for treating and preventing CDI.

## 1. Introduction

*Clostridioides difficile*, a Gram-positive, strictly anaerobic bacterium, is a major cause of nosocomial diarrhea in the USA and globally [1,2]. The incidence and severity of *C. difficile* infection (CDI) have increased dramatically since 2003 due to the emergence of NAP1/027 strains, also known as the hypervirulent strains of *C. difficile* [3,4]. Several factors are associated with an increased risk of CDI, such as the use of broad-spectrum antibiotics, increased age (>65 years), underlying chronic disease, and weakened immune defense [2,5,6,7]. *C. difficile* infects the host through the fecal–oral transmission of endospores, mainly affecting patients whose gut microbiota is depleted due to broad-spectrum antibiotics [2,7,8]. This disruption of normal colonic microbiota creates an environment conducive to the germination of spores into vegetative cells, which release toxin A and toxin B (some strains also produce binary toxin), damaging the intestinal epithelium [6,9,10]. Intestinal damage elicits a robust immune response in the colon, leading to a spectrum of clinical manifestations ranging from mild to severe diarrhea [6,9]. Severe cases may progress to life-threatening pseudomembranous colitis, which can result in fatal outcomes [9,11].

CDI is a significant public health concern in the USA and the EU. According to the 2017 CDC report, *C. difficile* causes around 200,000 cases annually, resulting in ~12,000 deaths in the U.S. alone [12]. The Centers for Disease Control and Prevention (CDC) list *C. difficile* as an urgent threat requiring aggressive preventative measures [12]. Further, CDI is estimated to cost USD 1 billion in associated U.S. healthcare spending. The impact of CDI extends beyond the USA, being a leading cause of hospital-associated diarrhea globally. In the European Economic Area’s acute care hospitals alone, the burden of healthcare-associated CDIs is estimated at 123,997 cases annually [13].

Asymptomatic carriage of toxigenic strains upon hospital admission is common, estimated at around 10 to 20% according to recent meta-analyses, significantly elevating the risk of developing CDI compared to non-colonized patients [14]. Although CDI typically occurs in hospitalized elderly patients who have been administered antibiotics, there has been a rise in the occurrence of CDI in the community among young adults who have not been exposed to antibiotics. Recent studies indicate the prevalence of community-acquired CDI, responsible for an estimated 20–27% of all cases, highlighting its substantial burden on healthcare systems [15].

*C. difficile* strains are categorized into various ribotypes, with hypervirulent ribotypes 027 and 078 being particularly noteworthy due to their association with severe disease presentations and high toxin production [16,17]. These strains produce several key toxins, including toxin A (TcdA), an enterotoxin that causes intestinal inflammation and fluid secretion by disrupting the intestinal epithelial barrier and attracting neutrophils, leading to further inflammation. Toxin B (TcdB), a cytotoxin, causes cell damage and apoptosis by disrupting the host cell cytoskeleton, resulting in cell death and significant colonic mucosa damage [18]. Additionally, *C. difficile* transferase (CDT), also known as binary toxin, disrupts the actin cytoskeleton, enhancing the pathogenicity of hypervirulent strains by promoting adherence to the intestinal epithelium and aiding in tissue invasion [19].

Multiple virulence factors are involved in the pathophysiology of CDI, but symptoms are mainly associated with the production of two exotoxins: toxin A and toxin B [9,20]. These toxins are part of the family of large clostridial toxins, characterized by high-molecular-weight proteins [21]. They act as glycosyltransferases, modifying specific GTPases, which disrupt the normal function of host cells [21,22]. Certain *C. difficile* strains, like PCR ribotypes 027 and 078, produce a third toxin called binary toxin (CDT) [19,23]. This binary toxin enhances the virulence of *C. difficile*. In this review, we concentrate on the roles of these three toxins in causing CDI symptoms and link their impact on cellular functions to the pathophysiology of CDI.

## 2. Overview of *C. difficile*’s Toxins

The signs and symptoms following CDI are influenced by various virulence factors, including the production of toxins and surface proteins [3,9,24]. The pathogenesis is primarily driven by the activity of toxins A and B, encoded in the pathogenicity locus of the *C. difficile* genome [25,26]. In the anaerobic colonic environment, *C. difficile* spores germinate into vegetative cells, which release toxins that are internalized by gut epithelial cells [9,10,20,27]. These toxins glucosylate small Rho proteins (Figure 1A), leading to actin-cytoskeleton dysregulation, impaired cellular signaling, and other detrimental cellular effects [20,28,29]. Rho proteins, targeted by TcdA and TcdB, constitute a group of small GTP-binding proteins of 21–25 kDa, belonging to the Ras superfamily [22,28]. The Rho family encompasses approximately 20 members, with notable representatives including RhoA, RhoB, RhoC, Rac1, Rac2, and Cdc42. These proteins serve as molecular switches involved in a multitude of signaling processes, acting as master regulators of the cytoskeleton and governing various cellular mobile functions such as phagocytosis, cellular trafficking, and migration [22,28]. Additionally, Rho proteins exert regulatory control over apoptosis and cell cycle progression [28]. One study unveils the structural mechanisms underlying the recognition and targeting of GTPases by *C. difficile* toxins, with TcdB variants categorized by substrate specificity [30]. Moreover, mutations impeding GTPase binding diminish TcdB toxicity, offering promising avenues for combating CDI. Similarly, the co-crystal structure of TcdA’s glucosyltransferase domain with RhoA sheds light on TcdA’s substrate selectivity, enhancing our understanding of clostridial toxin interactions [31].

CDI symptoms are further exacerbated by the host’s immune response, which includes an acute intestinal inflammatory response and neutrophil infiltration, causing additional damage to the epithelium [11,32]. Moreover, the compromised mucosal barrier, resulting from toxin-induced damage, facilitates the translocation of commensal bacteria into the lamina propria and systemic circulation, triggering robust inflammatory responses [33,34].

### 2.1. Toxin Structure

Toxins A (TcdA) and B (TcdB) share a broadly similar four-domain structure, exhibiting 47% amino acid identity (Figure 1B) [18,35,36,37]. Their N-terminal encompasses a glucosyltransferase domain (GTD), followed by a small cysteine protease domain that facilitates auto-processing for the GTD’s release. Adjacent to this is the Delivery and Receptor Binding Domain (DRBD), which contains a hydrophobic region and is believed to mediate the translocation of GTD from endocytic vesicles into the host cell cytoplasm [18]. Lastly, the C-terminal domain, also referred to as the C-terminal combined repetitive oligopeptide (CROPS) domain, is responsible for binding to various carbohydrates, potentially aiding in toxin attachment to the host cell surface [18,38].

### 2.2. Regulation of Toxin Production

Toxins A and B are encoded within the pathogenicity locus (PaLoc) of *C. difficile*’s genome (Figure 2A). These toxins bind to specific receptors on intestinal epithelial cells, enter the cells, and then glucosylate small Rho proteins [18,22,25]. This process leads to cell death, disrupts intestinal barrier function, and triggers an inflammatory response in the colon [11,33].

The pathogenicity locus (PaLoc) of *C. difficile*’s genome extends over a region of 19.6 kb [25,26]. This locus also contains three other proteins: tcdR, tcdE, and tcdC. TcdR functions as a sigma factor and is a positive regulator of toxin production [25]. In the absence of TcdR, purified *C. difficile* RNA polymerase fails to bind to the tcd promoter regions. However, TcdR and the RNA polymerase holoenzyme’s interactions enable transcriptional activation. Additionally, TcdR initiates its own promoter, establishing a positive feedback loop that regulates the whole PaLoc operon. TcdC serves as an anti-sigma factor, regulating toxin expression by modulating TcdR [25,26]. TcdE, on the other hand, acts as a holin-like protein responsible for toxin secretion. Holins, typical of double-stranded DNA phages, are membrane proteins crucial for host cell lysis. Interestingly, the tcdE open reading frame encompasses three translational start sites, resulting in three isoforms [26,39,40]. In hypervirulent strains of *C. difficile*, various combinations of these isoforms are implicated in both toxin release and cell death [40].

Certain pathogenic strains of *C. difficile* produce a third toxin called the binary toxin CDT, which is encoded by two genes, cdtA and cdtB [23]. Given the significance of binary toxin in disease pathogenesis, we will discuss it in a separate section below. These genes are located on a 6.2 kb chromosomal region called the Cdt locus, or CdtLoc, separate from the PaLoc region (Figure 2B). Within the CdtLoc exists a third gene, cdtR, responsible for encoding an orphan LytTR family response regulator. Notably, CdtR acts as a positive regulator of CDT production [18,19]. In epidemic ribotype 027 strains, CdtR also induces TcdA and TcdB production [18,19]. However, the environmental signals controlling CDT production and the mechanism of toxin secretion remain unclear.

### 2.3. Toxin Receptors and Toxin’s Effect on Disease Pathogenesis

Despite their structural similarities, the binding repertoires of TcdA and TcdB are independent from one another. Using genome-wide CRISPR-Cas9-mediated screens in the HeLa cell line, researchers have identified sulfated glycosaminoglycans as the primary receptors for toxin A [41,42]. In another study, through assays using murine fibroblasts and human colonocytes (Caco-2 cells), Low-Density Lipoprotein Receptor-Related Protein-1 (LRP1) was found to be an endocytic receptor for TcdA [18,43]. This finding is significant because TcdA must be internalized to exert its effects. Early studies on TcdA receptors showed TcdA-bound carbohydrate domains on the glucosidase enzyme sucrase–isomaltase in a rabbit small intestine, although this is not expressed in the colonic epithelium [44]. Using HT29 human colonic epithelial cells, Glycoprotein 96 was also identified to bind TcdA; however, it resides mainly in the endoplasmic reticulum and, therefore, is unlikely to be the primary TcdA receptor [45]. Chondroitin sulfate proteoglycan 4 (CSPG4) was identified as the initial receptor for TcdB through experiments conducted both in vitro (using HeLa, HT29, and HEK293T cell lines) and in vivo in mice [46,47]. Despite their high conservation, CSPG4 receptors are predominantly expressed on subepithelial mesenchymal cells rather than in the colonic epithelium, suggesting that CSPG4 may not be the primary receptor for TcdB [18,48]. Recently, Frizzled receptors 1, 2, and 7 have emerged as significant binding partners for TcdB. Experiments conducted with colonic organoids cultured from FZD protein-knockout mice have revealed that Frizzled receptors 1, 2, and 7 are significant binding partners for TcdB [49,50]. Their expression has been observed in the colonic epithelium, indicating their role in the interaction with TcdB. Furthermore, poliovirus receptor-like 3 (PVRL3) has been suggested as a potential receptor for TcdB. By employing a gene-trap insertional mutagenesis screen in Caco-2 and HeLa cell lines, potential receptors for TcdB such as poliovirus receptor-like 3 (PVRL3) have been identified [49,50]. However, the exact role of PVRL3 in TcdB-induced pathogenesis remains unclear, with uncertainty on its role in the pathogenesis of CDI. Recent studies have identified tissue factor pathway inhibitor (TFPI) as a receptor for toxin B (TcdB). TFPI is highly expressed in the intestinal glands, and recombinant TFPI has been shown to protect the colonic epithelium from TcdB-induced damage [51,52]. Notably, TcdB exhibits species selectivity in its interaction with TFPI, recognizing TFPI from chickens and, to a lesser extent, from mice, but not from humans, dogs, or cattle. These findings establish TFPI as a receptor for TcdB in colonic crypts [51,52].

The contributions of TcdA and TcdB in the pathogenesis of CDI have commonly been disputed. Based on animal studies, TcdA was thought to be the major virulence factor in infection, owing to a significant immune response to TcdA [53]. The initial concept suggesting that TcdA is the main driving factor in *C. difficile* pathogenesis was derived from experiments conducted with purified toxins. In the rabbit ileal loop and colon model of CDI, the addition of TcdA mounted the hallmark features of the disease, including increased mucosal permeability, inflammation, fluid secretion, and epithelial damage, while toxin B was found to have no effect [54,55]. In another study, the intragastric administration of TcdA to hamsters and mice resulted in inflammation, diarrhea, and eventual death, while TcdB showed no symptoms in these animals [56]. Surprisingly, in a hamster model of CDI, TcdB induced diarrhea and death when prior intestinal damage occurred or when sublethal doses of TcdA were co-administered [56]. This finding implies that prior damage caused by toxin A is crucial for the pathogenic impact of toxin B, suggesting a synergistic interaction between the two toxins in mediating toxic effects within the host.

In humans, toxin B plays a crucial role in inducing toxicity. Clinically relevant strains of *C. difficile* that produce TcdB but lack TcdA (A^−^B^+^) are associated with severe diarrhea in patients [57]. These A^−^B^+^ strains lead to a spectrum of clinical illnesses similar to A^+^B^+^
*C. difficile* strains in humans, ranging from mild diarrhea to pseudomembranous colitis and, in severe cases, death [57]. Interestingly, most A^−^B^+^ strains produce a modified TcdB variant capable of modifying Ras GTPases similar to TcdA [58]. This suggests that TcdB might perform glucosylation functions specific to TcdA when TcdA is absent. The virulence of A^−^B^+^ strains in infected individuals implies that TcdB alone is pathogenic in humans. Studies have provided further support for this observation by demonstrating that TcdB disrupts epithelial integrity and induces tissue damage in human colon explants [59,60]. Additionally, in a chimeric mouse model where human intestinal xenografts were transplanted into immunodeficient mice, TcdB also exhibited similar effects [59]. Moreover, recent phase III clinical trials have demonstrated that bezlotoxumab, a monoclonal antibody neutralizing TcdB, effectively reduces CDI recurrence in patients [61,62]. These findings underscore the pathogenic role of TcdB in *C. difficile* pathogenesis in humans.

### 2.4. Role of Toxin-Mediated Inflammation in CDI Outcomes

After the intestinal epithelium is damaged, TcdA and TcdB can penetrate the lamina propria and directly activate immune cells like dendritic cells and macrophages [6,53]. This activation leads to the release of cytokines and chemokines. The main inflammatory mediators released in response to toxin stimulation include IFNγ, TNF-α, IL-1β, IL-6, IL-8, and leukotrienes [33,53,63]. These mediators are crucial for recruiting other inflammatory cells, such as neutrophils, and coordinating the acute inflammatory response in the colon [11,64]. The role of toxin-induced inflammation in the pathophysiology of CDI is an active area of investigation. Research utilizing mouse and rat models of CDI suggests that toxin-triggered inflammatory reactions likely play a crucial role in causing *C. difficile*-associated colitis. In experiments involving mouse and rat ileal loops injected with toxin A, it has been demonstrated that drugs that inhibit the release of inflammatory mediators, like leukotriene B4, can diminish both neutrophil recruitment and mucosal damage [63].

The immune response mounted by toxin-induced damage serves as a defense mechanism for the host. However, it is important to note that the involvement of immune cells and cytokines in susceptibility to CDI is intricate and context-dependent, with multiple pathways and cell types contributing to the overall disease progression. For instance, research by Jarchum et al. revealed that depleting neutrophils in mice using anti-Gr1^+^ (Ly6G) antibodies exacerbates the severity and mortality of CDI [65]. This observation is consistent with clinical findings showing that neutropenia is associated with increased susceptibility to CDI and recurrent episodes in hospitalized leukemia and allogeneic hematopoietic stem cell transplant patients [66,67]. In contrast, another study demonstrated that inhibiting neutrophil infiltration in the colon through anti-CD18 (leukocyte adhesion molecule) treatment in rabbits resulted in reduced toxin A-induced enterotoxicity compared to the untreated group [68]. These studies emphasize the significance of employing diverse models of CDI to elucidate specific functions of immune cells and innate inflammatory responses in disease progression versus host defense mechanisms.

While inflammatory mediators play a crucial role in the initial defense against CDI, an uncontrolled response that results in the ongoing recruitment of inflammatory cells and prolonged intestinal inflammation is likely to be detrimental to the host. For instance, IL-23, a proinflammatory cytokine, is found at elevated levels in the colon during acute CDI in both mice and human patients [69]. Studies with mice lacking IL-23 signaling due to genetic knockout or monoclonal antibody-mediated neutralization have shown reduced CDI severity and improved survival compared to wild-type controls [69]. This suggests that IL-23 plays a role in mediating *C. difficile*-associated colitis. Similarly, another study found that higher levels of type 17 CD4^+^ T cells and their associated cytokines contributed to worse CDI outcomes [32]. Future studies are needed to uncover pathogenic inflammatory pathways and assess whether altering these signaling mechanisms could result in effective disease resolution.

*C. difficile* toxin B (TcdB) induces neurogenic inflammation by targeting pericytes and gut neurons via receptors such as chondroitin sulfate proteoglycan 4 (CSPG4) in pericytes and Frizzled receptors (FZD1, FZD2, and FZD7) in neurons [70]. Neurogenic inflammation occurs when sensory neurons release pro-inflammatory neuropeptides, causing rapid vasodilation and increased vascular permeability, which leads to plasma and immune cell extravasation into the tissue [71]. TcdB triggers the secretion of neuropeptide substance P (SP) and a calcitonin gene-related peptide (CGRP) from neurons and cytokines from pericytes, resulting in inflammation. The targeted delivery of the TcdB enzymatic domain, fused with a detoxified diphtheria toxin, into peptidergic sensory neurons expressing an exogenous diphtheria toxin receptor is sufficient to induce neurogenic inflammation, mirroring the intestinal pathophysiology seen in CDI [70].

## 3. Binary Toxin: *C. difficile* Transferase Toxin (CDT)

### 3.1. Overview of CDT

*C. difficile* produces another toxin referred to as *C. difficile* transferase toxin (CDT). This toxin is produced by approximately 5 to 30% of *C. difficile* strains [19,23]. It is commonly associated with hypervirulent strains of *C. difficile*, such as ribotype BI/NAP1/027, which are linked to heightened morbidity and mortality rates. CDT is part of the group of binary actin ADP-ribosylating toxins, which encompasses other toxins like *C. botulinum* C2 toxin, *C. perfringens* iota toxin, binary enterotoxin, *C. spiroforme* toxin, and the vegetative insecticidal proteins from *Bacillus cereus*/*Bacillus thuringiensis* [72]. These toxins comprise two distinct components. One component is responsible for receptor binding and facilitating toxin uptake, while the other component functions as an enzyme with ADP-ribosyltransferase activity [23,73].

### 3.2. Structure of CDT

CDTa, with a size of 48 kDa, is composed of two similar domains, suggesting an evolutionary duplication of genes [19,74]. Its N-terminal portion interacts with the binding component CDTb, while the ADP-ribosyltransferase activity resides in the C-terminal segment. Likewise, the binding component CDTb, measuring 98.8 kDa and consisting of 876 amino acids, bears a resemblance to the protective antigen found in anthrax toxin [72,74]. It is segmented into four domains: activation domain I (residues 1–257), responsible for initiating the process; domain II (residues 258–480), facilitating membrane insertion and pore formation; domain III (amino acids 481–591), promoting oligomerization; and domain IV (amino acids 592–876), involved in receptor binding. The activation of the binding component CDTb occurs through proteolytic cleavage of its N-terminal domain I, enabling oligomerization and heptamer formation [73,74]. This cleavage process may occur on the surface of target cells following receptor binding [19,73,74].

### 3.3. Receptors and Effect of CDT on Host Cells

CDT binds to the lipolysis-stimulated lipoprotein receptor (LSR), characterized by an extracellular immunoglobulin-like domain protein with an extended intracellular segment [75]. In experiments where the LSR gene was disrupted using CRISPR-Cas9 technology in the colon carcinoma cell line HCT116, it was observed that CDT binds to the LSR. The LSR is prominently expressed in intestinal, lung, liver, and kidney tissues [76,77]. Remarkably, this receptor is also recognized for its role in the clearance of low-density lipoproteins (LDLs) through a pathway independent of the LDL receptor [78]. The LSR plays a role in the formation of tricellular tight junctions, which are found at the points where three epithelial cells meet [79]. It recruits tricellulin, a protein associated with occludin, to these junctions [79]. After CDT binds to the LSR, the toxin–receptor complex undergoes endocytosis to access a low-pH compartment [75,80]. Subsequently, CDT translocates and undergoes refolding within the cytosol, facilitated by intracellular folding helper proteins like Hsp90 and peptidyl-prolyl cis-/trans-isomerase cyclophilin A [81,82]. Within the cytosol, CDT ADP ribosylates monomeric G-actin, consequently impeding actin polymerization [83]. This alteration in actin leads to modifications in the organization and dynamics of microtubules, which are elongated intracellular filaments composed of α/β-tubulin heterodimers [83,84]. These changes induce significant alterations in cell morphology, resulting in a loss of epithelial cell barrier functions and impacting various cellular functions dependent on actin polymerization, including phagocytosis, migration, secretion, and more [83,84].

CDT contributes to heightened bacterial virulence and severe disease by triggering inflammation and manipulating the host’s immune response during infection. In laboratory settings, CDT activates the NF-κB pathway in a murine macrophage-derived cell line [85]. Additionally, when combined with TcdA and TcdB, CDT increases the production of proinflammatory cytokines by murine bone marrow-derived dendritic cells [85]. Toll-like Receptor 2 (TLR2) signaling is necessary for the CDT-induced production of inflammatory cytokines by innate immune cells [85]. Eosinophils play a crucial role in host defense against acute CDI, and recent studies suggest that the TLR2-associated signaling pathway can induce eosinophil apoptosis, thereby suppressing a CDT-specific eosinophilic response in the colon [85,86].

## 4. Toxin-Based Strategies for Treating and Preventing *C. difficile* Infection

Most current therapies for the treatment of CDI are antibiotics [87,88]. Although these therapies do provide a good initial cure, unfortunately, antibiotics that can eradicate *C. difficile* bacteria can also impact the indigenous gut microbiota, and these therapies create an ideal niche for the growth of *C. difficile* bacteria, resulting in high rates of recurrent infections [87,88]. Therefore, *C. difficile* toxins are an attractive target for possible non-antibiotic therapies for the treatment of this infection. Initial studies in hamster models showed a high level of protection after immunization using formalin-inactivated *C. difficile* culture filtrates as antigens prior to infection [89]. Notably, protection against mortality and diarrheal disease was correlated with antibody responses to the various antigens tested [89]. A role for passive immunization with anti-toxin antibodies against the main *C. difficile* toxins (toxins A and B) showed promising results and provided protection against mortality, morbidity, and recurrent infections in hamsters [90,91]. In addition to these studies in animal models, data from CDI patients also support a role for antibodies against toxins A and B, where the serum levels of anti-toxin A and anti-toxin B antibodies correlate with protection from recurrent CDI [92,93]. These observations paved the way for the development of a new strategy for treating recurrent CDI, and worldwide clinical trials were conducted using monoclonal antibodies developed against toxin A and toxin B as a way to induce passive immunity for the prevention of recurrent CDI [61]. In MODIFY I and MODIFY II trials, CDI patients were given a single infusion of humanized monoclonal antibodies, either actoxumab (anti-toxin A antibody) or bezlotoxumab (anti-toxin B antibody) or a combination of the two, along with standard of care therapy for acute episodes of CDI [61]. There was a significant reduction in the rates of recurrent CDI in patients who received bezlotoxumab either alone or in combination with actoxumab [61]. However, despite the fact that clinical isolates of *C. difficile* make both toxins A and B, actoxumab treatment alone did not reduce recurrent CDI and did not provide any additive benefit when used in combination with bezlotoxumab [61]. These data led to the approval of bezlotoxumab as a treatment for recurrent CDI, and this medication remains a good option as an adjunctive therapy to reduce the risk of recurrent infection in high-risk individuals [94,95]. However, in individuals with a history of congestive heart failure, bezlotoxumab use has been associated with an increased risk of worsening cardiac disease, and therefore this medication is relatively contraindicated in such individuals [62].

Currently, anti-toxin antibody-based therapies have only been approved for recurrent CDI; however, non-antibiotic options are clearly needed for the initial episode of CDI. Vaccines are effective tools that can be targeted towards individuals who are at a high risk of developing CDI (e.g., those with advanced age, planned long-term antibiotic treatment for another infection, etc.) and for people in whom they are not expected to have a large impact on gut microbial communities. Further, since toxins are key drivers of CDI even during the first episode of CDI, they remain prime targets for developing novel CDI treatment strategies. As such, active immunization through the development of vaccines against *C. difficile* toxins is an attractive option that could add to our arsenal in the fight against CDI. In fact, some recent studies have shown promise in this regard, and individuals given a *C. difficile* vaccine candidate were reported to have the presence of toxin A and B neutralizing antibodies up to 48 months after three initial doses, with a further increase after a fourth dose [96]. While the bulk of the research on *C. difficile* has focused on toxins A and B, it is important to note that the clinically relevant strains of *C. difficile* also make a third toxin (CDT), which can contribute to CDI pathogenesis. Therefore, it will be important to consider all three toxins in the design of future vaccine candidates. In this regard, a new methodology that uses phage display to select high-affinity neutralizing antibodies to various regions of *C. difficile* toxins has great potential and has resulted in the identification of many potential targets in the CROP domain of *C. difficile* toxins [97]. Further exploration of these candidate molecules in animal models and clinical settings will be needed for the development of new toxin-targeted antibody tools for CDI treatment.

## 5. Conclusions

New research has enhanced our understanding of the structural and molecular mechanisms of TcdA, TcdB, and CDT, shedding light on their roles in CDI pathogenesis. While we have made progress in learning how these toxins affect target cells, there are still gaps in our understanding of their precise mechanisms in the disease’s progression. For instance, the mechanism by which *C. difficile* toxins deliver their effector domains to host cells remains unclear. Despite the identification of GTPase as a target for these toxins, further investigation is necessary to understand the determinants of GTPase substrate specificity and their implications for observed pathologies. Moreover, the intestinal epithelial cell receptor for TcdA has not yet been clearly identified, posing a significant question for future research. Additionally, the finding that TcdB can induce necrotic cell death independently of its glucosyltransferase activity introduces intriguing avenues for further study [98]. Questions also remain regarding the mechanisms and targets of binary toxin (CDT). Recent attention has been directed towards understanding how CDT contributes to disease pathogenesis. While CDT may enhance the pathogenicity of TcdA and TcdB, how it synergistically acts with other toxins to exacerbate disease severity remains unclear. Moreover, comprehending CDT’s involvement in pore formation and cargo translocation is essential for understanding its role in CDI progression. As mentioned earlier, CDT selectively induces the apoptosis of intestinal eosinophils in a TLR2-dependent manner, thereby increasing *C. difficile* virulence. However, the precise mechanism by which CDT achieves this and its potential effects on other immune cells remain poorly understood. Additionally, questions remain regarding the targets of CDT. Besides acting on the actin cytoskeleton, CDT also affects the microtubule system, leading to increased bacterial adhesion, colonization, and inflammation. A molecular understanding of this process would contribute to heightened bacterial virulence, and disease severity will be the focus of future research. Advancing our understanding of the toxin–host interaction pathway shows promise for developing more targeted therapies against this challenging pathogen.

## Figures and Tables

**Figure 1 toxins-16-00241-f001:**
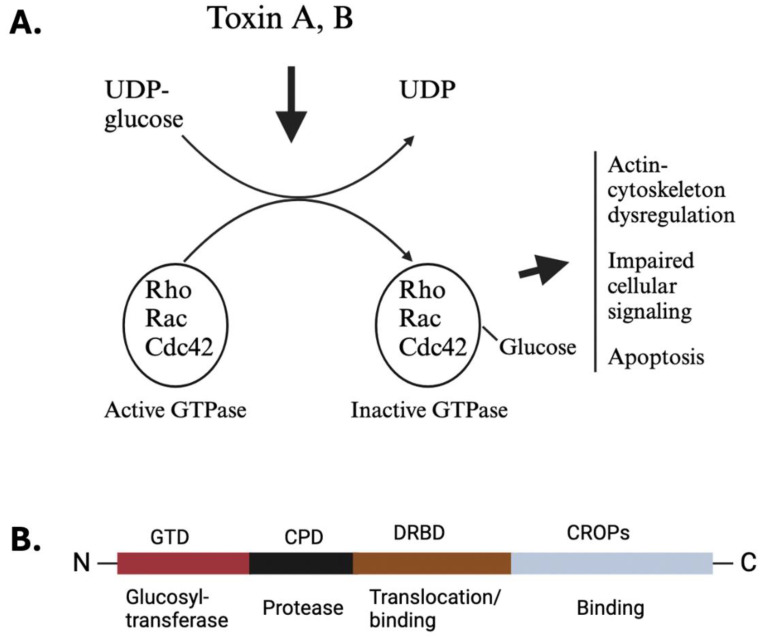
The deleterious effect of *C. difficile* toxins on host cells and their general structure. (**A**) The acidic environment inside the endosome triggers the release of the enzymatic glucosyltransferase domain situated at the N-terminus of toxins A and B. Once released, this glucosyltransferase enzyme inactivates small GTPases by adding glucose molecules. This disruption of GTPase function by the toxins leads to several deleterious cellular effects, such as disorganization of the actin cytoskeleton, weakening of tight junctions between cells, apoptosis, acute inflammation, and changes in cellular signaling pathways. (**B**) The schematics depict the basic layout of *C. difficile*’s toxins A and B, highlighting the four functional domains of TcdB. Starting at the N-terminus, there is the glucosyltransferase domain (GTD, red), followed by the inherent cysteine protease (black), the delivery and binding domain (DRBD, brown), and finally, at the C-terminus, the CROP domain (gray), which also plays a role in binding.

**Figure 2 toxins-16-00241-f002:**
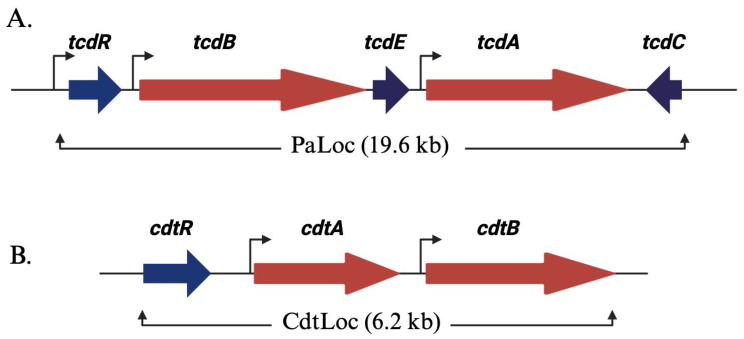
*C. difficile*’s toxin gene organization: (**A**) Illustration of the pathogenicity locus (PaLoc) in a schematic format. The toxin-encoding genes, tcdA and tcdB, are denoted by red arrows, while regulatory genes are represented in blue. The orientation of the arrows corresponds to the direction of transcription. TcdR exerts positive regulation on its own expression and that of tcdA and tcdB (highlighted by brown arrows). TcdC functions as an anti-sigma factor, exerting negative regulation on toxin expression by impeding TcdR’s function. TcdE plays a role in toxin secretion. (**B**) Depiction of the binary toxin locus (CdtLoc). Genes encoding for CDT, cdtA and cdtB, are illustrated in red. The regulatory gene, cdtR, is represented in blue. CdtR exerts positive regulation on the transcription of cdtA and cdtB.

## Data Availability

This review article did not create or analyze new data, and therefore, data sharing is not applicable to this article.

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
