# Peer review of "Clostridioides difficile Toxins: Host Cell Interactions and Their Role in Disease Pathogenesis"

_toxins, 2024, doi:10.3390/toxins16060241_

Round 1

Reviewer 1 Report

Comments and Suggestions for Authors

This is a review about C. difficile toxins. The topic is of interest and especially suitable for the journal "Toxins". Many exciting findings about this topic have been published recently. Thus, reviews are helpful to summarize important recent results and to describe new perspectives of the research field. Therefore, I recommend to improve the review by emphasizing recent important research results.

Special points

1. TFPI should be mentioned as a potential TcdB receptor.

2. The various clusters and types of toxins should be mentioned.

3. It should be clarified which results are only obtained from animal (mouse) studies and which are confirmed in humans.

4. The recently published “neurogenic inflammation” induced by C. difficile toxins should be mentioned.  

5. I am not sure whether the findings cited in reference 53 is of value for this review.

6. Is reference 13 correct? The publication is from 2015 and not from 2025.

7. Line 246: “Two identical domains”. Of course, the domains are not identical. They may be similar. The domains

8. Some therapeutic strategies are mentioned but not microbiota fecal transplantation, which is highly relevant.

9. Line 350: Please provide a reference.

Author Response

The authors greatly appreciate the reviewers' time and effort to review our article and provide valuable feedback. We have made every effort to incorporate all the suggested changes. We believe these revisions significantly enhance the strength and quality of this review paper. All changes in the revised manuscript are highlighted in red. Please do not hesitate to ask if you have further questions or comments. The authors are immensely grateful for your suggestions and comments. Please below see our replies to your comments:

TFPI should be mentioned as a potential TcdB receptor.

Reply: The authors thank the reviewer for this valuable comment. In the updated manuscript, TFPI has been incorporated with corresponding references (see lines 187 to 193).

The various clusters and types of toxins should be mentioned

Reply: The authors appreciate this comment. Due to the primary focus of this review on toxins, C. difficile clusters were not extensively discussed. The authors apologize for this oversight and have included a brief description in the revised version (updated lines 56 to 65).

It should be clarified which results are only obtained from animal (mouse) studies and which are confirmed in humans.

Reply: The authors thank the reviewers for their valuable insights. The identification of potent toxin receptors primarily involves genetic manipulation of in vitro cells, such as CRISPR-Cas9-mediated screens in the HeLa cell line. In this revised version, specific cell lines and techniques used for identifying toxin receptors are mentioned. For instance, in line 163, it is clearly stated that researchers identified sulfated glycosaminoglycans as the primary receptors for toxin A using genome-wide CRISPR-Cas9-mediated screens in the HeLa cell line. This updated information is also provided in lines 165, 170, 174, 179, 184, and 300.

The recently published “neurogenic inflammation” induced by C. difficile toxins should be mentioned.

Reply: The authors value this comment and consider it beneficial for providing updated information on the effects of C. difficile toxins to readers. Therefore, the role of C. difficile toxin in neurogenic inflammation has been incorporated into the revised manuscript along with references (updated lines 261 to 271).

I am not sure whether the findings cited in reference 53 is of value for this review.

Reply: The authors appreciate the insightful comments on reference 53 (PMID: 7738167). This study, conducted on in vitro human colonic epithelium, demonstrates toxin B's greater potency in damaging the epithelium compared to toxin A. We included this reference to complement our discussion on toxin B causing more severe CDI than toxin A in human patients, providing additional information for the reader. Therefore, we keep that reference.

Is reference 13 correct? The publication is from 2015 and not from 2025.

Reply: The author apologizes for this mistake regarding reference 13. The reviewer is correct. The publication is from 2015 and not from 2025. It is corrected in the revised version of the manuscript (Updated line 440)

Line 246: “Two identical domains”. Of course, the domains are not identical. They may be similar. The domains

Reply: The authors acknowledge this comment. The phrase "Two identical domains" in updated line 285 has been fixed to "Two similar domains." Thank you.

Some therapeutic strategies are mentioned but not microbiota fecal transplantation, which is highly relevant.

Reply: Thank you for your valuable feedback. We appreciate your comment and would like to address it. Our review primarily focuses on C. difficile toxins and therapeutic strategies stemming from understanding these toxins. Fecal Microbiota Transplantation (FMT) effectively treats recurrent CDI. Although its mechanism of action is not fully understood, FMT primarily works by clearing C. difficile from the colon, which is a pathogen control mechanism rather than a direct intervention on the toxins themselves. Given the scope and focus of our review, which is centered on toxin-based therapeutic strategies, we specifically chose not to include FMT. Please feel free to reach out if you have any further comments or questions. Thank you again for your time and insights.

Line 350: Please provide a reference

Reply: The authors appreciate this comment. Reference is added in the revised version of the manuscript. Update line 391; reference 98.

Reviewer 2 Report

Comments and Suggestions for Authors

The review titled "Clostridioides difficile Toxins: Host Cell Interactions and Their Role in Disease Pathogenesis" provides a brief overview of the structure and function of the toxins TcdA, TcdB, and CDT produced by Clostridioides difficile. It emphasizes the interactions between these toxins and host cells, offering valuable insights into potential interventions for Clostridioides difficile infection. Overall, this review is well organized.

Minor issues:

line 71-72: Please include two studies regarding the toxins in complex with their substrates in the citations (PMID: 34678063, PMID: 35637242).

line 99-100: An additional reference regarding the structure of TcdA should be included in the citations (PMID: 35292538).

line 156-163: Two recent studies have identified a novel receptor called TFPI for TcdB. This discovery should be discussed in the review and cited accordingly. (PMID: 36351897, PMID: 35303428).

line 246-247: Please refer CDT enzyme as CDTa.

Author Response

The authors greatly appreciate the reviewers' time and effort to review our article and provide valuable feedback. We have made every effort to incorporate all the suggested changes. We believe these revisions significantly enhance the strength and quality of this review paper. All changes in the revised manuscript are highlighted in red. Please do not hesitate to ask if you have further questions or comments. The authors are immensely grateful for your suggestions and comments. Please see below our replies to your comments:

line 71-72: Please include two studies regarding the toxins in complex with their substrates in the citations (PMID: 34678063, PMID: 35637242).

Reply: Thank you for your valuable feedback. We appreciate your comment. We have added these references along with a brief discussion in the revised manuscript (updated lines 90 to 95; references 30 and 31).

line 99-100: An additional reference regarding the structure of TcdA should be included in the citations (PMID: 35292538).

Reply: The authors appreciate this comment. In the revised manuscript, we have added this reference in the discussion on toxin structure (updated line 115; reference 37).

line 156-163: Two recent studies have identified a novel receptor called TFPI for TcdB. This discovery should be discussed in the review and cited accordingly. (PMID: 36351897, PMID: 35303428).

Reply: The authors thank the reviewer for this insightful comment. In the revised manuscript, a brief discussion including these two references has been added (lines 188 to 193; references 51 and 52).

line 246-247: Please refer CDT enzyme as CDTa.

Reply: The authors appreciate this comment. This has now been corrected to CDTa in the updated line 285.

Reviewer 3 Report

Comments and Suggestions for Authors

A review on the same topic containing most of the information given in this manuscript is already available (Given below). The authors have not cited this reference. Could the authors explain why this review is important when an article on this topic is available for the readers?

Buddle JE, Fagan RP. Pathogenicity and virulence of Clostridioides difficile. Virulence. 2023 Dec;14(1):2150452. doi: 10.1080/21505594.2022.2150452. 

Author Response

The authors greatly appreciate the reviewers' time and effort to review our article and provide valuable feedback. The authors are immensely grateful for your suggestions and comments. We greatly appreciate your feedback on our manuscript and the comparison you made between our review article and the review paper that you have mentioned (PMID: 36419222). Below, we address your questions and provide further clarification. We welcome your feedback and suggestions and are open to discussing any additional points you may have.

While both the review paper titled "Pathogenicity and virulence of Clostridioides difficile" and our review paper titled "Clostridioides difficile Toxins: Host Cell Interactions and Their Role in Disease Pathogenesis" share similar topics and touch upon overlapping ideas, they offer distinct perspectives and focus on different aspects of the role of toxins in Clostridioides difficile infection (CDI). Highlighting the differences between the two papers can provide readers with a more comprehensive understanding of the topic. Here are some ways in which they may differ:

1.     Depth of Coverage: The review article that the reviewer mentioned provides a broad overview of CDI, covering various aspects such as epidemiology, risk factors, clinical manifestations, diagnosis, drug-resistance and treatment options. Further, the paper provides a comprehensive overview of CDI pathogenesis, encompassing a broader range of virulence factors beyond toxins. It explores surface layer proteins, sporulation, and other mechanisms by which C. difficile establishes infection and causes disease. On the other hand, our review article delves deeper into specific structural and molecular mechanisms of toxins, including recent research findings and potential therapeutic strategies.

2.     Focus on Specific Toxins: While both papers likely discuss toxins like TcdA, TcdB, and CDT, our review article emphasis on the structural characteristics, receptor interactions, and downstream effects of these toxins at the molecular level. It specifically explores recent advances in understanding toxin-host interactions and toxin-based therapeutics for treating CDI.

3.     Research Gaps and Future Directions: In our review article, in conclusion, we dedicate more space to identifying gaps in current understanding and suggesting future research directions regarding toxin related aspect. We discuss unresolved questions regarding toxin mechanisms, host responses, guiding readers towards areas where further investigation is needed.

4.     Audience and Scope (Clinical Relevance vs. Basic Science): Review article that the reviewer mentioned may helpful to clinicians and healthcare professionals seeking practical insights into treating and drug-resistance aspect to CDI. In contrast, our review article may useful more to researchers and scientists interested in basic science research regarding C. difficile toxins and its mechanisms.

We acknowledge that both papers complement each other, offering valuable insights from different perspectives and levels of analysis. Again, we welcome your feedback and suggestions and are open to discussing any additional points you may have.

Round 2

Reviewer 3 Report

Comments and Suggestions for Authors

The manuscript can be accepted for publication.